# Incorporating Biodiversity into Biogeochemistry Models to Improve Prediction of Ecosystem Services in Temperate Grasslands: Review and Roadmap

**Marcel Van Oijen** [1,*], **Zoltán Barcza** [2,3,4], **Roberto Confalonieri** [5], **Panu Korhonen** [6], **György Kröel-Dulay** [7], **Eszter Lellei-Kovács** [7], **Gaëtan Louarn** [8], **Frédérique Louault** [9], **Raphaël Martin** [9], **Thibault Moulin** [10], **Ermes Movedi** [5], **Catherine Picon-Cochard** [9], **Susanne Rolinski** [11], **Nicolas Viovy** [12], **Stephen Björn Wirth** [11] and **Gianni Bellocchi** [9]

[1]  Centre for Ecology & Hydrology, Bush Estate, Penicuik EH26 0QB, UK
[2]  Department of Meteorology, Eötvös Loránd University, Pázmány P. s. 1/A, H-1117 Budapest, Hungary; zoltan.barcza@ttk.elte.hu
[3]  Excellence Center, Faculty of Science, Eötvös Loránd University, Brunszvik u. 2, H-2462 Martonvásár, Hungary
[4]  Faculty of Forestry and Wood Sciences, Czech University of Life Sciences Prague, Kamýcká 129, 165 21 Prague 6, Czech Republic
[5]  Università degli Studi di Milano, Cassandra Lab, via Celoria 2, 20133 Milan, Italy; Roberto.Confalonieri@Unimi.It (R.C.); ermes.movedi@unimi.it (E.M.)
[6]  Natural Resources Institute Finland, Halolantie 31A, 71750 Maaninka, Finland; panu.korhonen@luke.fi
[7]  Institute of Ecology and Botany, MTA Centre for Ecological Research, Alkotmány u. 2-4., H-2163 Vácrátót, Hungary; kroel-dulay.gyorgy@okologia.mta.hu (G.K.-D.); lellei-kovacs.eszter@okologia.mta.hu (E.L.-K.)
[8]  INRAE, URP3F, F-86600 Lusignan, France; gaetan.louarn@inrae.fr
[9]  Université Clermont Auvergne, INRAE, VetAgro Sup, UMR Ecosystème Prairial, 63000 Clermont-Ferrand, France; frederique.louault@inrae.fr (F.L.); raphael.martin@inrae.fr (R.M.); catherine.picon-cochard@inrae.fr (C.P.-C.); gianni.bellocchi@inrae.fr (G.B.)
[10]  Agroscope, Agroecology and Environment, Reckenholzstrasse 191, CH-8046 Zurich, Switzerland; thibault.moulin@agroscope.admin.ch
[11]  Potsdam Institute for Climate Impact Research, Member of the Leibniz Association, P.O. Box 60 12 03, 14412 Potsdam, Germany; rolinski@pik-potsdam.de (S.R.); wirth@pik-potsdam.de (S.W.)
[12]  Laboratoire des Sciences du Climat et de l'Environnement, LSCE/IPSL, CEA-CNRS-UVSQ, Université Paris-Saclay, F-91191 Gif-sur-Yvette, France; Nicolas.Viovy@Lsce.Ipsl.Fr
\*  Correspondence: mvano@ceh.ac.uk

**Abstract:** Multi-species grasslands are reservoirs of biodiversity and provide multiple ecosystem services, including fodder production and carbon sequestration. The provision of these services depends on the control exerted on the biogeochemistry and plant diversity of the system by the interplay of biotic and abiotic factors, e.g., grazing or mowing intensity. Biogeochemical models incorporate a mechanistic view of the functioning of grasslands and provide a sound basis for studying the underlying processes. However, in these models, the simulation of biogeochemical cycles is generally not coupled to simulation of plant species dynamics, which leads to considerable uncertainty about the quality of predictions. Ecological models, on the other hand, do account for biodiversity with approaches adopted from plant demography, but without linking the dynamics of plant species to the biogeochemical processes occurring at the community level, and this hampers the models' capacity to assess resilience against abiotic stresses such as drought and nutrient limitation. While setting out the state-of-the-art developments of biogeochemical and ecological modelling, we explore and highlight the role of plant diversity in the regulation of the ecosystem processes underlying the ecosystems services provided by multi-species grasslands. An extensive literature and model survey was carried out with an emphasis on technically advanced models reconciling

biogeochemistry and biodiversity, which are readily applicable to managed grasslands in temperate latitudes. We propose a roadmap of promising developments in modelling.

**Keywords:** biodiversity; biogeochemistry; ecosystem services; environmental change; experiments vs. non-intrusive studies; functional types; process-based modelling; species; traits

## 1. Introduction

### 1.1. Grasslands as Major Providers of Ecosystem Services

Grasslands cover a large fraction of the land surface in temperate climatic zones. In Europe, about 40% of agricultural area is occupied by grasslands [1], and globally the value is 70% [2]. Grasslands in temperate zones are characterized mostly by how they are maintained (e.g., by grazing or mowing) as well as by local environmental conditions. Different types of grasslands can emerge, ranging from extensive grasslands (which are important for nature conservation) to intensively managed pastures or meadows (mostly exploited for agricultural production on fertile soils). Most grasslands are managed to produce feed for animals, and the increasing demand for high quality animal products that comply with health, ethical and environmental standards requires sustainable grassland management, particularly in the face of global changes. In this context, beyond environment and management drivers, a third dimension of important drivers for grassland functioning and dynamics emerges: the diversity of plant species [3]. The relations between species diversity and ecosystem functioning can strongly influence grassland biogeochemistry, with expectations about positive effects of high species richness. In particular, there are societal and policy expectations about the role of species-rich grasslands for the delivery of provisioning, regulating, supporting and cultural ecosystem services (ES, https://cices.eu/resources). Grasslands are not only the core of forage production worldwide but they also provide multiple additional ES such as carbon sequestration, erosion control and habitat for pollinators and other fauna [4]. Grassland species richness promotes carbon storage in plant and soil pools [5]—a robust store compared to other systems [6], which offers an opportunity to restore or even enhance existing carbon reservoirs [7]. However, European grasslands, as currently managed, are a small net source of greenhouse gases (GHG) if all farm processes are taken into account [8–10].

Grasslands also provide important cultural ES, which cover non-material services that contribute to mental and physical well-being, e.g., as part of the visual appeal of landscapes (the English Lake District) or as the locus of desired landscape elements (windmills in the Netherlands). These cultural ES are not amenable to the kinds of experimentation and modelling that we examine here, and will not be considered further.

Supporting services are those needed to produce all other ES because they ensure the proper functioning of the ecosystem. These services include the major biogeochemical cycles (water, carbon, etc.), soil conservation, community structure and primary production. We here adopt the perspective that biodiversity supports and modulates the provision of most ES (Figure 1) [11]. This role of biodiversity is well established: the services that can be provided by grasslands depend in part on the diversity of their floristic composition [12] and their high biodiversity consisting not only of plants, but also of mammals, arthropods and microorganisms [13,14]. The importance of plant biodiversity is emphasized because it is closely linked to soil health and functioning, due to its benefits for soil microorganisms and other fauna [15]. Plant biodiversity thus supports ES but it is also recognized as an ecological and evolutionary insurance (after [16]) thanks to the stabilizing effect of species diversity on aggregate ecosystem properties through fluctuations of component species (e.g., phenotypic changes, [17]). Traditional management practices like grazing, mowing and fertilization are major drivers of vegetation dynamics and species distribution, but the combined effects of management practices and environmental changes on biodiversity-mediated ES remain unclear (e.g., [18]). We know,

for instance, that plant diversity would make grasslands more resilient to hazards and extreme weather events (such as prolonged droughts, e.g., [19,20]) and would be able to stabilize forage production and maintain overall ecosystem services [21]. Most native grassland communities contain a high diversity of drought-tolerant grassland species [22]. In areas with severe environmental stresses, the establishment of pasture communities including plant species with different functional characteristics can help ensure that ecological stability is a key adaptation measure to climate change [23]. Overall, plant diversity can act as a safeguard of ecosystem functioning for heat stress [24] and flooding events [25]. It is therefore essential to preserve these open spaces in order to maintain their biodiversity and the associated services, but also to study them to better appreciate their evolution under different constraints [26]. While there is mounting evidence that biodiversity increases the stability of ecosystem processes in changing environments, the mechanisms that underlie these effects in grasslands are still poorly understood [27]. Thus, determining the nature and value of the goods and services that grasslands can provide under different environmental and management conditions remains an open field of research that has not yet been fully exploited (e.g., [28]). Changes in plant diversity reflect the evolution of two main factors (after [29]): environmental conditions outside human control (pedo-climate) and management practices that aim to optimize the conditions for ES-provision (mowing, fertilization, grazing, etc.). These factors show complex interactions [30] and it thus remains difficult to accurately predict the dynamics of biodiversity and ES-provision in multi-species grasslands [31].

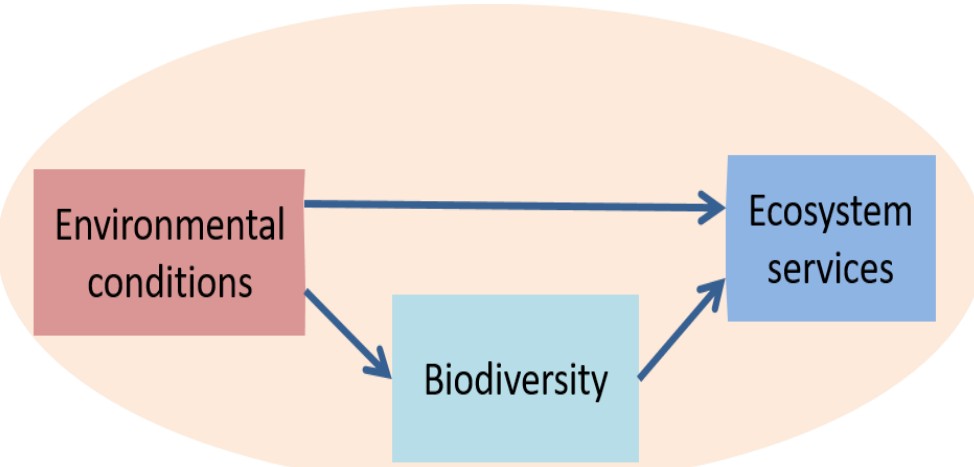

**Figure 1.** Direct and indirect relationships between environment (climate, soil, management), biodiversity and ecosystem services. Biodiversity supports and modulates the ecosystem services that can be provided under given environmental conditions.

### 1.2. Using Models to Explain and Predict Grassland ES

In the environmental sciences, *biogeochemistry models* (BGMs) are often used to quantify the land-surface exchange of GHG, and in agronomy, BGMs are used to simulate growth and productivity. BGMs are dynamic process-based models of the pools and fluxes of carbon, nutrients and water through soil-plant-atmosphere systems (Figure 2). Many of the ES provided by grasslands, including productivity, are related to fluxes of mass and energy, so they can be modelled using BGMs. BGMs thus can help explain observed spatial variation of grassland ES, as well as predicting ES responses to environmental change. However, most BGMs do not represent biodiversity or are restricted to a small range of plant functional types (PFTs). This limits their ability to explain or predict the impact of environmental changes on those ES whose response is modulated by biodiversity. Note that in this paper, we mostly use a restricted definition of biodiversity, as only pertaining to the higher plant species that are present in a grassland community. We do allow for various biodiversity metrics,

including species richness, diversity of traits, functional types and relative abundance, which reflect different features of biodiversity [32].

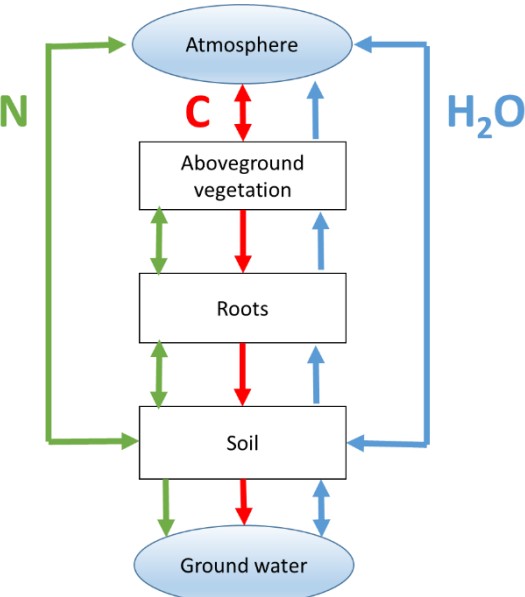

**Figure 2.** The typical structure of a biogeochemistry model (BGM).

The dynamics of biodiversity and its relationships with ecosystem functioning have been modelled too, but with *ecological models* rather than with BGMs [33]. These ecological models tend to be relatively simple, mathematically elegant models that focus on biotic interactions whilst ignoring or strongly simplifying biophysical processes. Due to this, the interactions between biogeochemistry and biodiversity, and their interdependent effects on ES, have not been represented well in either BGMs or ecological models. Therefore, model improvements are needed, which we aim to elucidate in this paper.

The study of the relationships between grassland biodiversity and ES requires converging ecological and biogeochemical approaches that account for agricultural practices. The wealth of knowledge gained from empirical studies is the ground on which this position paper is based, while capitalizing on the views of modelers and experimentalists in grassland ecology. The integration of these views in a conceptual framework relies on our understanding of the controls exerted by environmental and management drivers on the provision of biodiversity-mediated ES. Models should address how environmental and management changes affect grassland biodiversity, how such biodiversity is linked to ES, and how changes in biodiversity and ecosystem functioning feed back to the microclimate and soil conditions.

### 1.3. Goals and Outline of This Paper

In this paper, we focus on managed and unmanaged grasslands in temperate climates and, as mentioned, our measure of biodiversity is restricted to higher plant diversity. We shall mainly consider those ES that have been studied intensively, such as grassland productivity, carbon sequestration and GHG emissions. Our environmental drivers of interest include climate variables and farming practices, including grazing or mowing and fertilization. We mainly consider studies at the field scale, not landscape or regional scale.

Tackling plant diversity modelling to unravel the relationships between this diversity and the functioning of grasslands can provide answers of direct interest to farm managers and policy makers to maintain or improve certain ES. For example, identifying mixtures to optimize biomass production is a field of study that aims to enhance the potential of multi-species grasslands towards high-quality

livestock production. In this frame, the mobilization of models simulating several species can help guide management towards the maintenance of balanced mixtures under changing conditions [34]. Otherwise, there is a need to characterize changes in the botanical composition of grasslands during winter because, particularly in high latitudes and mountainous areas, they may have significant spring and summer effects on productivity and nutritional quality [35]. In Mediterranean climates, on the other hand, the dominance of annual species in vegetation (although mixed with perennials) requires an adaptive plant-seeding strategy that is responsive to the unpredictability of hydrological conditions [36].

As noted, BGMs are simplified models in that plant diversity is, at best, reduced to a simplistic schematization of interactions between species or PFTs (see the review by Ma et al. [37]). It is therefore important to improve the representation of plant diversity effects on the C-N cycles as well as the water cycle and energy balance. We explore how generic BGMs can be extended in a way that ensures consistency between the growth dynamics of each represented plant species and the biophysical and biogeochemical flows in the community.

With this aim of model improvement in mind, our goals in this paper are as follows:

- We review the current understanding of the impact of grassland biodiversity on ES, and the drivers of biodiversity itself (Section 2).
- We review the state of the art of models for grassland biogeochemistry, and of models for the dynamics of biodiversity (Section 3). We identify the main differences between the two modelling approaches.
- We discuss how BGMs can be modified to simulate both the dynamics of biodiversity itself and the impacts of biodiversity on ES (Section 4).
- We discuss our findings and propose a roadmap for further model development and data use (Section 5).

## 2. Why Consider Biodiversity in Grassland Models?

In this section, we give a brief overview of the empirical evidence for the role of biodiversity in grasslands: how biodiversity affects ES, and how biodiversity itself responds to external drivers. This is followed by a brief discussion of the relative merits of different kinds of experimental and monitoring studies and their implications for modelling.

### 2.1. Impacts of Biodiversity on ES

We first summarize the empirical evidence for the impact of biodiversity on ES-provision by grasslands. Quantifying the impact of biodiversity on ES is not straightforward. Many environmental drivers affect both biodiversity and ES, so relations between biodiversity and ES can be correlative rather than causal. To facilitate the detection of causal pathways, many experiments have been performed with grasslands where different species compositions were imposed by the researcher and resulting differences in sensitivity to environmental perturbation were quantified. Key findings are:

- Biodiversity (observed species richness) affects the extent to which drought reduces growth and soil respiration [38]. More generally, biodiversity (species number in manipulated grassland communities) increases resistance to various types of climatic event: dry or wet, moderate or extreme, short- or long-term [39]. In marginal grasslands (e.g., on dry soils in the Mediterranean), high biodiversity may extend the growing season. This shortens the period in which soils are bare and therefore protects against soil loss by erosion. Despite the fact that high biodiversity tends to stabilize productivity by reducing climate sensitivity [39], there are examples where biodiversity reduces the stability of ecosystem functioning in response to extreme events [40].
- High biodiversity tends to lead to higher aboveground biomass, especially when species are from different functional types [41]. In grazed grasslands, high biodiversity not only improves

productivity, but also grass quality and milk production [42], although the improvement may be absent under already highly productive conditions [40].

- High biodiversity in intensively managed grasslands may suppress weeds [43], but a modest increase in conventional grassland plant diversity with legumes and forbs improves pollination and therefore productivity [44]. In unfertilized grasslands, the presence of legumes increases productivity and soil organic matter, and improves soil texture [45]. The soil improvement also increases resilience against drought.
- Permanent monocultures will eventually decline in productivity due to insect herbivory or pathogen load [46].
- High species richness reduces root decomposition by increasing root C:N ratios, except for legumes [47], with grassland species having widely varying turnover rates [48]. However, high biodiversity (in mixtures that were compared to monocultures) does tend to stimulate soil microbial biomass and soil respiration [49], and it strongly increases carbon sequestration [50,51]. Overall, N-cycling processes are stimulated less by biodiversity than C-cycling [50], except when legumes are added to the mixture [52].

### 2.2. Impacts of Environmental Change on Biodiversity

We now consider the dynamics of biodiversity itself. The literature provides many examples of impacts of environmental change on the biodiversity of grasslands. The main drivers that have been examined are management effects (intensity, frequency and timing of cutting or grazing, weed management, choice of sown species, fertilization), climatic effects, and to a lesser extent, the effects of pests and diseases. Key findings are:

- In mixtures subjected to an imposed extreme drought, grass species common to wetter soils (high value of the Ellenberg F index) suffered most senescence and mortality [53]. Legume species suffered more than grasses, irrespective of their Ellenberg value. Recovery was also better in grasses than in the N-fixers.
- Experimental warming tends to decrease species richness, but its impact depends strongly on the specific ecosystem [54].
- Biodiversity reduction in a Californian grassland was due to decreasing winter rain, and not due to changes in grazing, fire, N-deposition, or invasive species [55]. Losses especially concerned native annual forb species with traits indicative of low drought tolerance. In contrast, 13-year long manipulation of temperature and rainfall in infertile grasslands had only a minor effect on biodiversity (species richness and relative abundance of growth forms) and productivity with the exception of reduction from chronic summer drought [56].
- In a Mediterranean silvo-pastoral system, grazing intensity did not affect species richness but it did affect species composition [57]. In the same system, grassland biodiversity was shown to be enhanced by the habitat variation induced by the presence of the trees [58].
- A review of herbaceous systems worldwide showed that nitrogen fertilization tends to reduce plant species richness, whereas irrigation mostly has little effect [59]. Fertilization reduces plant species richness of mountain grasslands the most where summers are cool, where mowing is carried out, and overall where biomass has been increased most by the fertilization [60]. The effects of phosphorus are generally less evident than those of nitrogen [61].
- Grazing tends to be more beneficial for biodiversity than mowing [62]. Johansen et al. [63] observed that abandonment of grazing led to decline in species diversity in Norwegian semi-natural grasslands but the effects varied with climate and soil conditions. However, grazing affects biodiversity nonlinearly: minor grazing increases biodiversity, overgrazing reduces it [64–66], while also the timing and the grazing species are important [67].
- In short-rotation (3–4 years) grasslands, the diversity of plant species is initiated by farmers deciding what combination of species is sown and how growth of weeds is suppressed, but weed

proportions generally do increase, thereby decreasing sward productivity when they replace high yielding grass, forb and legume varieties [68].

### 2.3. What Can We Learn from Experiments and Field Observations?

Experiments have generally shown positive impacts of biodiversity on ES. Two likely high-level explanations for this are *selection* and *complementarity*, as proposed by Loreau and Hector [69]: in mixtures of many species there is a greater likelihood of well-adapted high-biomass species being present, and mixtures may better exploit available resources, although the degree of complementarity depends on the types of species that are present [70,71]. However, detailed mechanistic explanations are still lacking. The impacts of biodiversity must be realized through the ecophysiological processes in the grassland, but the key mechanisms have generally not been identified in great detail [50]. There is some scattered information. Functional composition was a stronger determinant of stress responses than species richness in an alpine area: forbs in managed grassland plots had higher specific leaf area, N-demand and growth rate than forbs in abandoned plots, which explained both the greater drought sensitivity and greater post-drought recovery (N-uptake, growth) of the managed grassland [72]. It also seems increasingly likely that the effects of higher plant biodiversity are partly mediated via co-evolving soil microbial composition [73], with soils under low management intensity being dominated by fungi vs. bacteria-dominated intensively managed soils, which affects nutrient availability to the plants.

In a very interesting study, Liu et al. [74] used structural equation modelling to identify the likely causal mechanisms linking fertilization, biodiversity and ES in a Chinese grassland (Figure 3). They found evidence for both direct and biodiversity-mediated effects of N-addition on two ecosystem services: aboveground net primary production and ecosystem stability. Their study is a rare example where the general scheme of Figure 1 was clarified in detail.

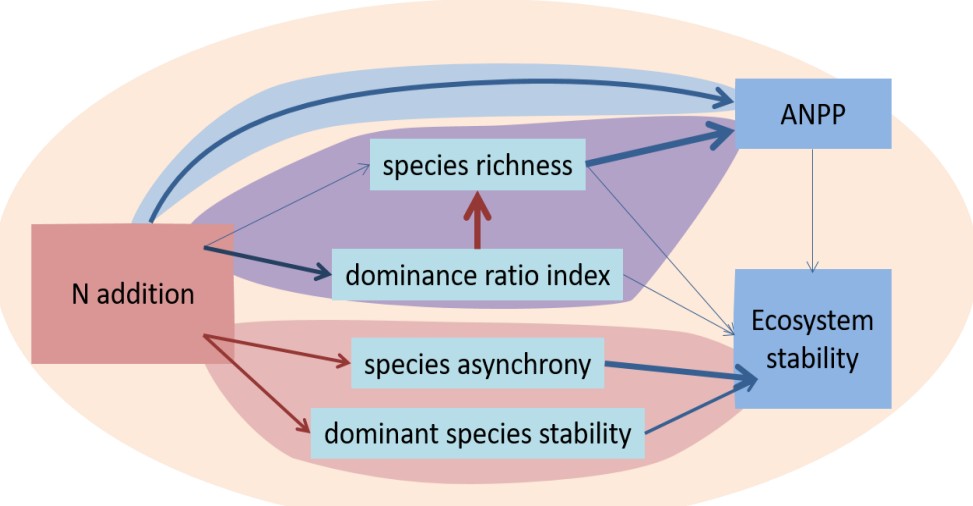

**Figure 3.** Modified after Liu et al. [74]: An example of causal relationships in fertilized grassland. Blue arrows: positive effects; red arrows: negative effects. Arrow width indicates strength of effect. An odd number of red arrows in a pathway implies a negative effect. For example, high asynchrony between species stabilizes the system, but N-addition reduces asynchrony, so that particular pathway reduces stability.

Generally, empirical information is difficult to interpret and always contingent on study conditions. There have been studies of grasslands both in controlled environments and under field conditions. In studies where the effects of biodiversity were being studied, species composition was either manipulated or contrasting field plots were selected and observed in situ or in mesocosms. In some of the studies where biodiversity was initialized by the researcher, biodiversity was kept fixed through

weeding [50], whereas others allowed invasion of other species. Both short- and long-term studies have been carried out. The types of grasslands also vary: from nearly unmanaged systems that are close to equilibrium, to intensively managed systems that are always in disequilibrium. When interpreting the literature on grasslands, we have to take all these different study conditions into account.

There are conflicting results from (1) experiments in which humans impose the levels of biodiversity and (2) observational studies where different levels of biodiversity result from differences in natural processes (coevolution of habitat diversity and species diversity [75]) and management. In (1), productivity tends to increase with biodiversity, whereas in (2) highest productivity is seen in intensively managed grasslands with low biodiversity.

The impact of the set-up of the experiments has been studied by various authors:

- Grace and colleagues [76] showed that the influence of biodiversity on productivity was very small in mature natural grasslands.
- Veen et al. [77] and Weisser et al. [50] showed that results from weeded experiments may differ from non-weeded experiments. See also Kardol et al. [78] who concluded that their species-removal experiment (non-weeded) allowed for better low-diversity performance than weeded species-addition experiments.
- Gruner et al. [54] found that the experimental set-up affected the response of biodiversity to warming.
- Long-term effects of biodiversity are often stronger than short-term effects [70], emphasizing the importance of long-term ecological research. In particular, below-ground processes respond slowly to manipulations of biodiversity [50].

Obviously, the goals of the different kinds of study vary. Studies with a high degree of manipulation, e.g., those in which the researcher selects and combines different grassland species, generally aim to identify the causes underlying relationships between biodiversity and sward performance. Such studies are not intended to identify the drivers of biodiversity itself. More observational field studies, on the other hand, aim to quantify the impacts of environmental change on both ES and biodiversity. This means that the two study types support modelling in different ways. The manipulation experiments are valuable for model design: deciding what mechanisms to represent in the model. The observational studies are valuable for model calibration and for testing predictive capacity (Figure 4). The models themselves, in their turn, may assist in explaining the conflicting results from the two types of studies and, in particular, the apparent mismatch in their findings concerning the biodiversity-productivity relationship. We summarize the roles of the different empirical and modelling approaches in Figure 4.

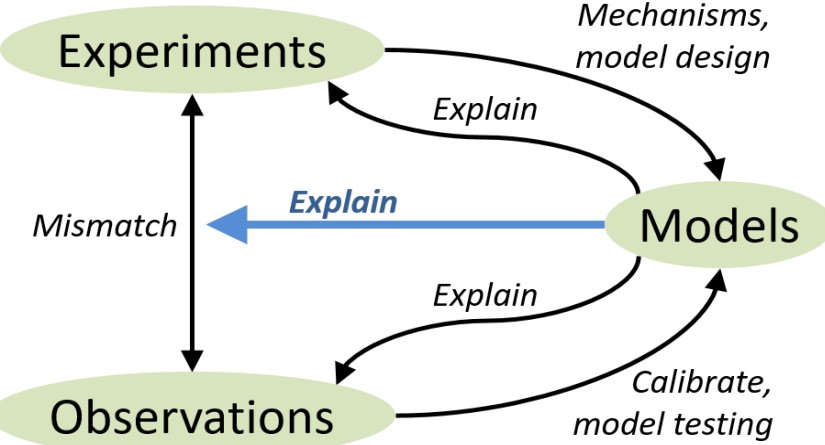

**Figure 4.** Relationships among experiments, non-intrusive observations and modelling. See main text for discussion.

## 3. Modelling the Biogeochemistry and Biodiversity of Grasslands

In this section, we briefly discuss the general structure of both biogeochemistry models and ecological models for biodiversity, focusing on applications to grasslands. We analyze the differences between the two model types, expanding on an earlier comparison by Van Oijen et al. [33].

### 3.1. Biogeochemistry Models (BGMs)

BGMs, whose overall structure was shown in Figure 2, typically simulate the flows of carbon, nitrogen and water between pools in the atmosphere, aboveground vegetation, roots, soil and, in some cases, also ground water (e.g., [79]). Importantly for the purposes of this paper, many of these flows simulated by BGMs represent ES, e.g., productivity, C-sequestration, reduction of GHG emissions and N-leaching. The BGMs can simulate the response of these ES to changes in climate, fertilization, atmospheric [$CO_2$] and pollutants, etc.

The overall modelling approach of BGMs is mechanistic, so processes are dynamically simulated with control mechanisms that determine their rates. To represent the processes and their controls, BGMs also represent morphology (e.g., LAI, rooting depth), allocation strategies, phenology and, in some models, the dynamics of soil microbial biomass [79]. The models simulate seasonal effects through seasonal variation in weather variables and their impact on the phenological status of the plants. BGMs typically are point-support, one-dimensional models that are used to simulate processes in a homogeneous area, such as a grassland field. They can be applied over larger areas if information about spatial variability of driver variables is available. Boundary conditions for the simulated dynamics are the initial conditions of soil and atmosphere, and prescribed time series of weather variables and human intervention for management and harvesting, but biotic interactions are not represented in detail. All this means that most BGMs can simulate feedbacks from resource-limitation but not the dynamics of biodiversity and its impacts on ecosystem functioning. For example, there are now multiple grassland BGMs that can simulate the impact of nitrogen availability on grassland production and quality in monocultures [80–82], but that capacity has not yet been demonstrated for mixtures.

Some BGMs do include multiple species albeit without simulating the dynamics of sward composition. An example is ModVege [83,84], which assumes a constant assemblage of species or PFTs. The lack of species dynamics may be considered a weakness of typical BGMs, given the empirical evidence for the impacts of biodiversity on ES that we reviewed in Section 2. There are now also dynamic modelling studies that demonstrate this. For example, Fitton et al. [85] modelled grass-legume interaction with two models: with and without competition. They showed that above 30% legume cover, only the model with competition provided realistic results. In Section 4, we shall discuss different recent approaches to introducing biodiversity dynamically into BGMs.

### 3.2. Models for Biodiversity

We now summarize approaches to modelling plant diversity dynamics, mainly from the ecological literature. Methods proposed in recent years range from minimal ecological models—often expressed in elegant mathematics—to more comprehensive integrated or individual-based computer models. The models represent interactions between herbaceous species or PFTs by simulating population dynamical processes (birth, death, migration) and competition. In most cases, the models are developed to address general ecological questions and choose to minimize the level of physical and chemical detail, in particular for the soils, to preserve the models' general applicability, and to limit the number of parameters that need to be estimated.

Theoretical models of plant competition, non-exhaustively reviewed by Gillet [86], describe the dynamics of multi-species coexistence as dependent on the consumption of resources. They are useful tools to explore hypotheses on how competition determines the structure and dynamics of herbaceous communities, e.g., the conditions of coexistence of populations, plant succession, and patterns of species abundances. The competition rules of the models tend to be relatively simple. Typical examples

are the models of Tilman et al. [87], which explain the positive correlation between diversity and productivity by community optimization of the consumption of resources. These models also illustrate the *tradeoff-based niche theory* [88], where plant biomass is regulated by resource dynamics, and they illustrate the *principle of competitive exclusion*, which states that two different species cannot coexist if only a single resource is limiting [89]. Moulin et al. [90] refined this approach by coupling the competition dynamics to ecophysiological processes, which allowed them to simulate both seasonal growth dynamics and the persistence of more species than there were essential resources. Schwinning and Parsons [91] and Loreau et al. [92] showed that the stability of multi-species communities might arise from large-scale spatial heterogeneity. Recently de Mazancourt et al. [93] used stochastic differential equations adapted from the Lotka-Volterra equations to explain multi-species coexistence in communities from the stochasticity of environmental and demographic variables. Indirect competition processes, defined by the tolerance to external disturbances of dominant species or PFTs, were shown in the GraS model [94] to account for vegetation succession. Succession has also been modelled using the state-and-transition formulation of Westoby et al. [95].

Individual-based grassland models (IBMs) are flexible tools for simulating the dynamics of biodiversity. They can explicitly simulate high numbers of species or functional groups with above- and below-ground intra- and inter-specific competition [96], as was done in the IBC grass model [97], which was able to simulate communities of many PFTs over hundred years. May et al. [98] used individual-based models that accounted for shoot and root competition processes to simulate grazing reversal, whereby grazing increases biodiversity in productive environments but decreases it in unproductive sites. Milchunas et al. [99] also illustrated with their modelling study how moderate grazing tended to increase species richness compared to ungrazed swards (see Fox [100] for a rejection of the Intermediate Disturbance Hypothesis on theoretical and practical grounds). The model EcoHyD [101] described the cover dynamics of a large number of PFTs of semi-arid grasslands and savannas but from hydrological processes only, and with growth rate determined by just two parameters (maximum water uptake rate and water-use efficiency) rather than simulating the biogeochemistry of carbon and nutrient flows. Bittebiere et al. [102] showed that the predictions of IBMs for grassland composition were highly sensitive to the formulation of competitive interactions, in part because in such models resources are considered as abstract quantities, not bound to conservative recycling.

In the modelling of biodiversity dynamics, we can distinguish approaches that focus on the abundances of PFTs, species or traits, with increasing attention for trait-based approaches in recent years [103]. Laughlin et al. [104,105] proposed a Bayesian hierarchical approach whereby environmental conditions favor certain trait compositions, which in turn determine likely species compositions. The plasticity of the trait distribution response to the environment depends on the available genetic variability of traits, and Van Oijen et al. [106] showed how Bayesian methods can be used to infer, from macroscopic measurements, the genetic variance of grass genotype traits that are difficult to observe directly.

*3.3. The Main Differences between BGMs and Models for Biodiversity-ES Relations*

As we showed above, researchers interested in the causes and consequences of biodiversity have generally taken other modelling approaches than modelers of biogeochemistry. The different modelling choices hamper integrated modelling. We can distinguish differences in model structure, mathematical formulation, data-use, and spatial scale.

- *Model structure*. The simulated state variables in BGMs tend to be pools of carbon, nitrogen and water in soils, vegetation and atmosphere, whereas ecological models tend to focus on population dynamical properties such as the abundance of PFTs, species, traits or individuals (and their size-age distribution). The processes that are modelled are biogeochemical/physiological and demographic, respectively. BGMs focus on abiotic environmental drivers whereas biodiversity models usually focus on biotic interactions. For BGMs, traits are static inputs (i.e., model parameters) while for biodiversity models they are dynamic state variables.

- *Mathematical formulation*. BMGs are never fully analytical models: trajectories over time of state variables must be numerically derived by computer modelling. Biodiversity models can be analytically solvable systems of differential equations, but even when they are computer models, their time step of calculation tends to be much longer than the daily or sub-daily time step of most BGMs. Moreover, biodiversity computer models tend to be discrete-event agent-based (or even individual-based) models in contrast to the 'big-leaf' continuum BGMs.
- *Data-use*. BGMs require detailed information on mainly abiotic conditions (weather, atmospheric [$CO_2$], N-deposition, soil properties) as drivers for the flows of carbon, nitrogen and water, whereas biodiversity models predominantly need biotic information, such as the initial age-size distribution of organisms or the frequency distribution of traits.
- *Spatial scale*. BGMs tend to be one-dimensional models that are assumed to be applicable to spatially homogeneous fields, whereas the representation of space in biodiversity models may be poorer in the vertical direction (no leaf or soil layers) but richer horizontally, even extending to simulating heterogeneous landscapes rather than fields (see also Figure 5).

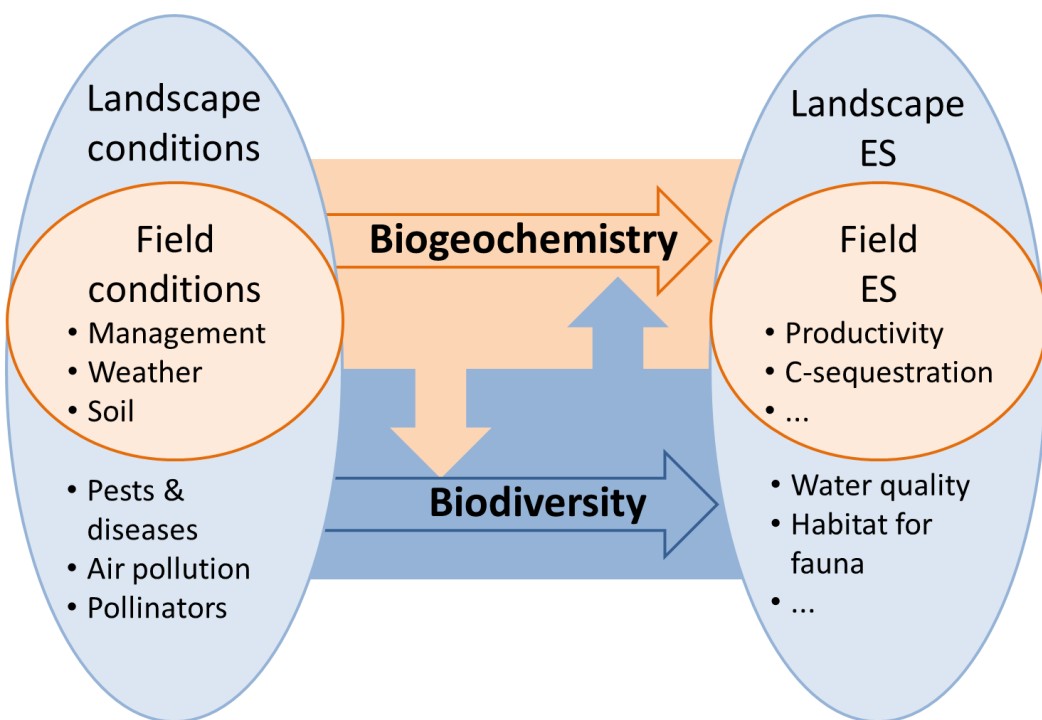

**Figure 5.** Biogeochemistry and biodiversity at field and landscape scale, with scale-dependent environmental drivers and provision of ecosystem services (ES).

## 4. Modifying Existing BGMs to Simulate the Impacts and Dynamics of Biodiversity

We now turn to the question of how existing BGMs can be modified to simulate the interaction between biogeochemistry and biodiversity, and their impact on ES. We distinguish three different approaches that range from simple and empirical to complex and mechanistic.

### 4.1. Representing Biodiversity as a Constant Metric

The simplest way to incorporate biodiversity in an existing BGM is to introduce a biodiversity-parameter that quantifies the dimension of biodiversity that is deemed most relevant to the purposes of the BGM. The parameter could be species richness or any other metric. A minor extension of this idea would be to include multiple biodiversity-parameters (in other words, a biodiversity-vector) quantifying not just species richness but also functional diversity or species evenness. Doležal et

al. [107] proposed a metric to quantify the level of asynchrony of each species in a mixture: average correlation of its biomass to that of the rest. The biodiversity-parameter(s) could then operate as a modifier of the BGM's original parameters, processes or outputs. We have not found examples of this approach in the literature, but there are precedents in other vegetation modelling, such as the nitrogen availability parameter of Tilman's models, or the empirical site fertility rating (FR) parameter of forest model 3-PG [108]. Moreover, empirical research [93] has shown that biomass variability over time, which is an output of BGMs, can be modelled as a direct function of species synchrony and dominance, which suggests that a biodiversity-vector could be an effective BGM-output modifier.

Apart from the non-mechanistic modelling of biodiversity-impacts, an obvious weakness of the parametric approach is that biodiversity is deemed constant, which is often not a correct assumption (see Section 2), except for short periods without major disturbances. This needs to be weighed against its ease of implementation.

### 4.2. Representing Multiple Species or PFTs without Simulating Competition

We now investigate methods for extending BGMs with dynamic simulation of biodiversity that do not rely on implementing competition for resources. In order to simulate environmentally determined succession of vegetation types, some BGMs integrate a Dynamic Vegetation Model (DVM). This type of model is often used to simulate shifts in potential vegetation due to global change (in which case they are referred to as DGVMs), but examples can also be found at the field scale, e.g., by Thornley [109] who presented a grass-legume mixture model, and by Confalonieri [110] with his "CoSMo" approach which will be elucidated below.

Classically, DVMs adopt methods from biogeography and compute—at each time step—the levels of "suitability" of the prevailing environmental conditions for existing and alternative vegetation types. By this means, the abundance of a species increases when environmental or management conditions improve for itself or worsen for other species in the community. Competition is thus not explicitly simulated: there is no calculation of interspecific differences in resource use, the changes in population sizes follow from differences in suitability. In fact, in this approach model structure regarding biogeochemistry is not changed at all: the same processes are simulated as before. However, the calculation of process rates will now depend on the relative abundances of the different plant species or PFTs where their parameter values differ. In short, the impact of biodiversity becomes an emergent property of community dynamics.

In the case of the CoSMo-approach [110,111], the suitability functions are based on a suite of variables: (i) management events (grazing, cutting), (ii) weather variables, (iii) resource availability (water and nitrogen), and (iv) relationships between state and rate variables of the community (e.g., LAI and rooting depth), and on species-specific parameters (functional traits). At each time step, after the abundances have been updated following the changes in suitability, the BGM needs to calculate the impact of the new species composition on the biogeochemical processes. This is done by abundance-weighted averaging of either parameters or process rates. In the first approach (Figure 6, Case 1), at each time step, we drive a single BGM-calculation forward using the average plant parameter vector. This leads to updated rates and states of the BGM after which the cycle of {suitability calculation → abundance updating → parameter averaging → state updating → suitability calculation} is repeated for the next time step. In the second approach (Figure 6, Case 2), at each time step, we compute the process rates separately for each individual species (thereby multiplying the number of required BGM-calculations) and calculate the abundance-weighted average process rates, and use these mean rates to update the states.

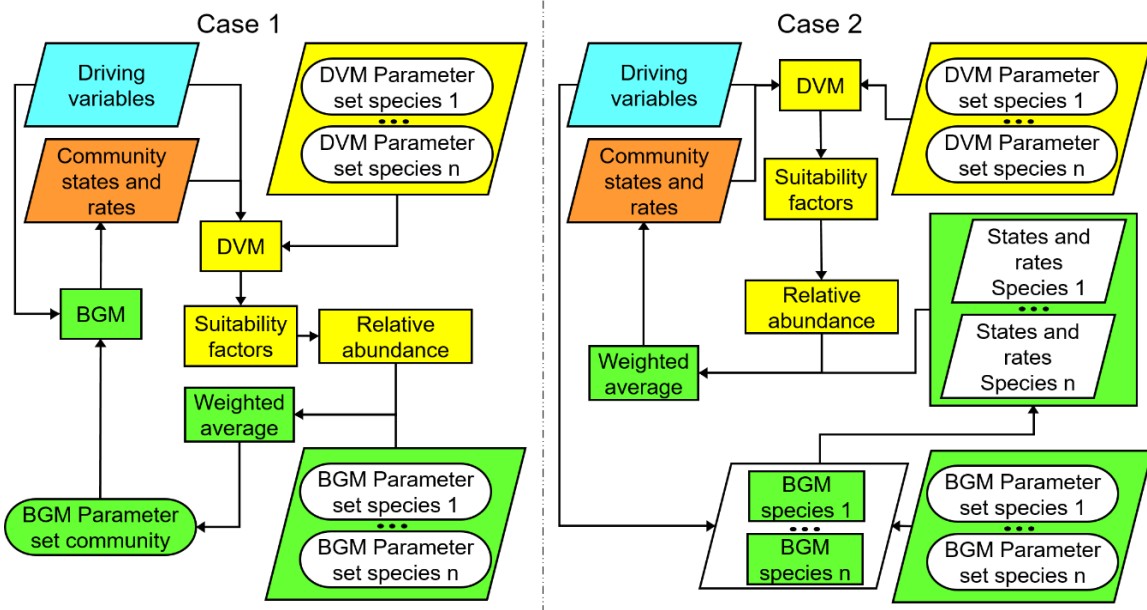

**Figure 6.** Two ways to implement the CoSMo approach. Case 1: parameter averaging. Case 2: process rate averaging. Background colors indicate different parts of the modelling framework. Cyan: external drivers. Yellow: dynamic vegetation modelling. Green: biogeochemistry modelling. Orange: state equations.

Both CoSMo-approaches can be easily implemented by coupling an already existing BGM with a DVM which computes the relative abundances of the species at each time step. The first approach does not increase the computational time and is easier to implement. However, it is based on the assumption that traits (i.e., model parameters) in a mixture can be treated as additive. The second approach allows more realistic, non-linear impacts of trait variation but increases the computational demand by a factor equal to the number of species. Both approaches have the same demand for information about traits, which need to be specified for each individual species in the community.

### 4.3. Representing Multiple Competing Species or PFTs

Explicit, process-based simulation of competition comes closest to the mechanistic logic underpinning BGMs. In this approach, it is not necessary to simplify the interactions between species or PFTs, so there is no need to use an external DVM for calculation of the suitability of plant species to the environmental conditions [110]. Instead, competition for resources (light, water, and nutrients) is simulated within the BGM, with the individual species or PFTs being parameterized separately. This logic inevitably complicates the parameterization of the BGM and may therefore lead to increased output uncertainty if trait information is limited. Nevertheless, as the ultimate aim is to construct grassland models that realistically simulate the responses of processes to long-term environmental change, such consideration of biodiversity may be necessary (see e.g., [85]).

When introducing competition into a BGM, the phenology, physiology, and fluxes of carbon, nitrogen and water through each competing species or PFT will be calculated separately. Therefore, not just the number of parameters but also the number of state variables and rates will increase, and most modules of the original BGM will need modification or multiplication. To minimize parameter uncertainty and model structural complexity, most attempts at modelling competition in BGMs have sought to simplify parts of the modelling. Competition-enabled models can be categorized by the complexity of process-representation (competition for light only, or also for water and nutrients, impacts of disturbances and stress-factors, etc.), or by the degree of spatialization (ignoring spatial heterogeneity, using simple geometry models for patches, individual-based modelling; see [112]). IBMs

may apply the gap concept originally used in forest modelling [96,113], but may also bridge the gap to BGMs by mechanistically modelling competition for soil resources [114,115].

The simplest models that include competition simulate just two plant species or PFTs. Common examples of this are models for crop-weed [116], crop-tree [117,118], and grass-legume interaction [109, 119–121]. Comparisons of some existing multi-species models designed for intercropping, agroforestry and forestry are provided by Malezieux et al. [112]. Focusing specifically on grassland models, Taubert et al. [96] and Snow et al. [2] provide overviews of different approaches. Snow et al. [2] showed that models tend to differ less in their implementation of competition for light than in their simulation of belowground processes and their representation of spatial heterogeneity. However, BGMs exist that simulate the impact of litter substrate quality on soil microbial biomass and the impact of the microbial biomass on organic matter decomposition and nutrient availability [79]. Such models could be expanded with the capability to not only simulate microbial biomass but also—in an aggregated way—microbial biodiversity which is known to vary with plant species diversity through differences in litter quality [49] and which in turn affects nutrient availability to the plants [73]. In some of the grassland models, belowground processes are not implemented at all, so they can only simulate competition for light. The most complex methods include functional-structural modelling of the root-shoot system and its implications for light interception, water use and nutrient uptake [122–124].

We see that multi-species grassland models (e.g., [90,125,126]) are slowly becoming more prevalent. However, among the 13 grassland models reviewed by Taubert et al. [96], only two considered more than three species or PFTs. This included the GRASSMIND model, which simulates several grass species competing for resources dependent on species-specific traits and allometric relations. In the vast majority of such competition-enabled models, the trait-values for each species or PFT are considered to be constant in time and space. Community properties can thus only change if species- or PFT-abundances change. A different route is taken for the individual-based model LPJmL-FIT [113]: for several traits, each individual is given a randomly drawn value from the leaf economics spectrum [127]. Therefore, the community of surviving plants results in a trait distribution adapted to current environmental conditions. Another example of trait plasticity is the GEMINI model [126,128], which optimizes plant traits of average individuals in combination with trade-offs as manifested in the leaf economics spectrum [127]. Competition driven by differences in physiological and morphological plant characteristics results in trait combinations adapted to the environmental conditions. This trait plasticity feedback allows a reaction of the plant community to environmental changes in terms of biodiversity and ecosystem services.

## 5. Discussion

### 5.1. The Need and Scope for Introducing Biodiversity into BGMs

Our literature review of Section 2 confirmed that there is clear evidence that biodiversity may affect the provision of ES by grasslands. Average productivity, its response to environmental change, and grass quality can all vary with the composition of the sward. Grassland biodiversity itself varies over time and space as well. Taken together, the evidence implies a need for introducing biodiversity—both its dynamics and impacts—into BGMs, if the models are to be able to predict the impact of environmental change on ES-provision by grasslands. Our paper thus elaborates on two of the challenges for modelling European grasslands under climate change that were identified by Kipling et al. [129], i.e., "Modelling multi-species swards" and "Modelling the provision of ecosystem services".

However, there are still considerable knowledge gaps, which may hamper the development and testing of models. The literature on grassland ES is dominated by studies of productivity; impacts of biodiversity on other ES are not nearly as well studied. This can be explained from both the traditional dominance of economic considerations, and from the fact that many ES such as regulating and supporting ES (filtering of ammonia and nitrate, water regulation, erosion prevention, GHG emissions

reduction, carbon sequestration) have been more difficult to measure. Many of these particular ES are expressions of the biogeochemical cycles in grasslands, and are therefore amenable to simulation by BGMs, but the knowledge base needs to be expanded to do so reliably.

We showed that there is a trade-off between the realism of an experiment and its capacity to detect causality. Experiments under highly unrealistic conditions—where the researcher imposes and maintains a desired species mixture—can be highly valuable for the study of causality, i.e., detecting the ecological mechanisms and feedbacks that constrain grassland dynamics. Such manipulation experiments may thereby provide key information for designing the causal structure of a model. However, we also need to know what biodiversity does in agricultural and natural grasslands, in which the response to environmental factors (such as nitrogen availability) may seem opposite to what is observed in the experiments. Such non-intrusive observational studies provide the response patterns that a BGM needs to be able to explain and predict, to be of use.

Despite the availability of manipulation studies, our understanding of the biotic and abiotic controls on grassland processes remains incomplete. When reviewing the results of the 15-year Jena grassland experiment, Weisser and colleagues [50] concluded that the mechanisms underlying observed positive effects of biodiversity on grassland ES are not well known, and that likely several mechanisms are acting simultaneously. This lack of understanding hampers the identification of optimal model structure, but this can also be seen as a motivation for comparative mechanistic modelling: designing differently structured complex mechanistic models as a means of quantifying the relative importance of different proposed mechanisms.

The data requirements of BGMs extended with biodiversity are considerable because the properties of all components of the community must be specified. For some model parameters, modelers can take advantage of trait databases such as TRY [130]) or other databases such as e-FLORA-sys (http://eflorasys.univ-lorraine.fr) and LEDA (https://uol.de/en/landeco/research/leda/data-files). Trait databases may not always be useful because the expression of traits can be highly plastic, with complex dependence on overall community composition [50]. The bottom-up approach of measuring parameters should thus be complemented by the top-down approach of model inversion (data-assimilation) in which Bayesian methods are used to derive probability distributions for parameters from measurements of fluxes and other model output variables [106]. This in turn requires the availability of reliable benchmarking data from sites where biogeochemistry and biodiversity have been studied in concert.

Overall, we conclude that model design, calibration and testing would all benefit from a broader empirical knowledge base, involving both manipulation studies and non-intrusive observations (Figure 4).

## 5.2. Reconciling Current BGMs and Models for Biodiversity

We compared typical BGMs and biodiversity models, and found considerable differences in model structure, mathematical formulation, data-use, and spatial scale. Due to these differences, the two model types cannot be linked in a simple way. Overall, BGMs tend to be more complicated than biodiversity models, so a natural approach is to start from a BGM and add key elements from biodiversity modelling rather than the other way around. We identified three general approaches for this, ranging from introducing biodiversity as a scalar or vector metric to simulating biodiversity as a set of state variables, without or with explicit simulation of competition. The latter approach especially will increase the complexity of the model considerably. The greater complexity may however facilitate parameter estimation if it involves replacing empirical competition coefficients (with site-specific values) with measurable traits.

In modelling, there is always a trade-off between realism (complexity) and computational efficiency. In current BGMs, one common simplification is to define PFTs that are supposed to represent the typical behavior of specific plant groups. The properties of PFTs, in the models, are supposed to be the same everywhere, so the traits of each PFT are generally represented by a set of fixed parameters. However, this is an important limitation because it restricts spatial heterogeneity to abiotic factors and it disallows

seasonal or long-term adaptation of traits to any changes in environmental conditions. Recently, several ways have been explored to alleviate these limitations. One of these is to define empirical relationships between environmental conditions and the different traits [103,131], using recent efforts to build global trait databases such as the aforementioned TRY [130]. Traits then become environmental drivers rather than model parameters. However, this approach is constrained by the availability of trait data from which to derive robust relationships. Moreover, it is limited to traits that can be observed, which is not the case for all of the traits represented in models. This can partly be solved by inverse modelling of the environment-trait relationships from FLUXNET or satellite data as proposed by Peaucelle et al. [132], who followed the general approach of Bayesian parameter calibration [106,133]. Another more theoretical approach is to consider the fact that plants tend to optimize their trait-values within the constraints of taxa-specific overall plant growth strategies [56,134]. An example of such optimization theory is the coordination hypothesis of photosynthesis [135], which identifies the optimal leaf nitrogen concentration at which the rates of carboxylation and Rubisco regeneration are co-limiting equally. This leaf nitrogen concentration delivers maximum nitrogen-use efficiency and is indeed observed over a large spectrum of conditions and plant types. The theory thus allows defining maximum photosynthetic rate as a function of environmental conditions rather than as a constant PFT-specific parameter [136]. Another example is the leaf economics spectrum [127], which states that there is a trade-off between leaf nitrogen concentration, leaf longevity and specific leaf area: thin leaves have a high N concentration and low longevity whereas thick leaves have low N concentration and high longevity. Combining coordination of photosynthesis and the leaf economics spectrum should thus allow three major plant traits to vary in the model directly with environmental conditions, although the extent to which the presence of competitors may change the location of the optimum for each species needs to be ascertained. For plant traits that vary less continuously between species (e.g., architectural parameters, which strongly affect resource competition), a stronger link between field experiments and the developing high-throughput phenotyping capacities [137] may also contribute to improving trait databases, thus allowing more robust calibration of biodiversity-based models.

*5.3. A Roadmap for Future Model Development*

We argued for the need to incorporate biodiversity into BGMs to improve prediction of ES in grasslands, discussed difficulties and identified possible solutions. In this closing section, we outline promising research directions. We are unable to define the optimal way forward, so this is just a roadmap of possible routes that can be taken.

*Mechanistic modelling: nearby destinations.*

- It will always be useful to have a choice of models, with different levels of complexity, and with different domains of applicability. The three different approaches to joint modelling of biogeochemistry and biodiversity that we distinguished in Section 3 are not mutually exclusive, but can be explored in parallel. Each approach comes with its own set of strengths and weaknesses, which can be assessed by frequent model comparisons against common data.
- It will be worthwhile to keep pursuing the various methods for managing model complexity that we described above (under "Reconciling Current BGMs and Models for Biodiversity"), in particular the identification of further constraints to trait-trait and trait-environment relationships. However, a trade-off between ease of implementation and parameterization on the one hand, and realistic representation of mechanisms on the other, will remain unavoidable.
- The biogeochemical modelling will be of immediate benefit to our understanding of biodiversity-ES relationships if it explores ways to reconcile the apparent mismatch between responses of ES to manipulated biodiversity and naturally evolved biodiversity (see Figure 4).
- It will also be important to use biodiversity-representing BGMs to explore why certain responses to environmental change are seen in some grasslands but not in others, as in the examples provided

by De Boeck et al. [40] where high biodiversity did not necessarily lead to greater stability of ecosystem functioning in response to extreme events.

- As BGMs are dynamic models, they can, in principle, be employed to explain how the initial response of grasslands to disturbances (resistance phase) differs from the long-term response (recovery phase), given that biodiversity can affect the two phases differently [72].

*Mechanistic modelling: long-range destinations.*

- In the long term, future biogeochemistry-biodiversity modelling may aim for more ambitious applications than those documented so far. This includes more comprehensive representation of environmental drivers (e.g., phosphorus, weeds, pests and diseases) and more detail in the representation of biodiversity itself along the different dimensions of PFTs, species and traits.
- Most current BGMs have limited or no representation of spatial processes. They tend to be one-dimensional models that can be run for different locations, with site-specific environmental drivers, but true spatial processes such as species migration and hydrology are generally not represented [138]. Given the fact that the dynamics of biodiversity are only partly determined by within-field processes, future models for biodiversity and biogeochemistry may have to operate at the landscape-scale, as depicted in Figure 5. Applying process-based models to larger spatial scales requires that the interplay of species dynamics and grassland functioning is analyzed at local and regional scales, and assessed in virtual landscapes with heterogeneous soil, climate, management and natural disturbance and stress factors.

*Data collection and benchmarking.*

- Model development cannot proceed without supporting data, and data analysis is hampered when reliable models are not available. The future modelling developments outlined above should thus go together with continuing increases in the quantity and diversity of data. We argued above for the parallel development of multiple modelling approaches, but this development should frequently be re-anchored in reality by comparing all models against rich benchmarking data sets, that cover multiple different ES rather than just productivity.
- Benchmarking data should be collected for the many different production situations that grasslands experience: potential growth, water-limitation, nutrient-limitation, weeds, pests, diseases, grazing, mowing. The data should cover extreme events (abiotic, biotic) as well as chronic stress conditions, in particular those that are expected to become more common in the future. Biodiversity may act to stabilize biogeochemical fluxes under extreme conditions, and BGMs need data to test their capacity to account for this.
- Remote sensing and eddy covariance measurements can be collected to assess the prevalence of biodiversity and its impact on GHG over wider areas. Eddy covariance towers have generally been placed in spatially homogeneous landscapes, to facilitate data interpretation despite variable wind directions, but this is no longer a necessity. Levy et al. [139] showed how Bayesian inference can be used to derive a spatial map of vegetation properties around a single measurement tower. A caveat is that eddy covariance measurements on grasslands tend to be unreliable during grazing events because carbon losses from animal respiration are not registered [140].
- The literature shows much evidence for an important role of biodiversity in grasslands, but uncertainties remain because of differences in environmental conditions between studies. Meta-analyses of available data should, for example, not conflate the impacts of experimentally manipulated vs. naturally evolved biodiversity. A Bayesian hierarchical approach to meta-analysis could be used to account for the interactions [141], which would prepare the data for use in model development.
- Overall, we advocate that data keep being collected in both biodiversity-manipulation experiments, and in monitoring studies where biodiversity is not imposed by the researcher (Figure 4). The first

study type will provide data that elucidate causal pathways and can thus be used in the design of model structure. The second study type provides data from the actual grasslands for which our models need to be calibrated to issue reliable forecasts.

*Hybrid mechanistic-probabilistic modelling.*

- We have focused here on mechanistic modelling of biogeochemistry and its relationship with biodiversity in order to predict ES under new conditions, and to explain observations of ES. However, we may want to predict a more comprehensive suite of grassland ES than is customary or even possible for biogeochemical modelling. For ES that are not outputs of BGMs, modelers could assess whether those ES are predictable from variables that the models do simulate. Aesthetic appeal, for example, may be a function of biodiversity. These added relationships are not mechanistic and may not be robust, but they can be implemented as conditional probability distributions to account for uncertainty.
- More generally, for practical application of our models and for ease of uncertainty quantification, we may want to summarize the input-output relationships of our biodiversity-enhanced BGMs in the form of graphical models (probabilistic networks), which will facilitate uncertainty and risk analysis [138].

**Author Contributions:** G.B. and M.V.O. designed the study, M.V.O. led the writing, Z.B., R.C., P.K., G.K.-D., E.L.-K., G.L., F.L., R.M., T.M., E.M., C.P.-C., S.R., N.V. and S.B.W. all contributed equally to the text. All authors have read and agreed to the published version of the manuscript.

**Funding:** This work was supported by the Agence Nationale de la Recherche of the French government through the program 'Investissements d'Avenir' (16-IDEX-0001 CAP 20-25). M.V.O acknowledges support from the Research Council of Norway through project MYR (No. 281109), the Environment & Rural Affairs Monitoring and Modelling Programme (ERAMMP) (Welsh Government Contract C210/2016/2017) and the UK Centre for Ecology & Hydrology (Project: 06297). Z.B. was supported by the Széchenyi 2020 programme, the European Regional Development Fund, the Hungarian Government (GINOP-2.3.2-15-2016-00028) and by OP RDE (CZ.02.1.01/0.0/0.0/16_019/0000803). P.K. was supported through project DivCSA (the Academy of Finland, decision no. 316215). S.R. and S.W. acknowledge financial support from the project Climasteppe (BMBF under grant 01DJ18012).

**Acknowledgments:** This work was undertaken under the auspices of the project MODIPRAS (Modelling relationships between species diversity, the functioning of grassland systems and their ability to deliver ecosystem services) of the INRA meta-programme ECOSERV (Ecosystem services). We thank three anonymous reviewers of the manuscript for their helpful suggestions.

**Conflicts of Interest:** The authors declare no conflicts of interest.

## Abbreviations

BGM = Biogeochemistry model; DVM = Dynamic vegetation model; ES = Ecosystem services; GHG = Greenhouse gases; IBM = Individual-based model; PFT = Plant functional type.

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
