# Peer review of "Incorporating Biodiversity into Biogeochemistry Models to Improve Prediction of Ecosystem Services in Temperate Grasslands: Review and Roadmap"

_agronomy, doi:10.3390/agronomy10020259_

Round 1

Reviewer 1 Report

As a review, this is very nice work.  It captures much of the relevant literature, lays out where we are and provides some thoughts about moving to a more comprehensive understanding of grasslands through coupled biodiversity and biogeochemical modeling.

While I think it is publishable as it is, it does fall short in that it provides very little detail on exactly what biogeochemical processes can be coupled to biodiversity or how that can be done with specific examples.  The authors lists some processes and services, but they didn't create any concrete examples.  For instance, what might a biogeochemical model of belowground N transformations look like when coupled to plant diversity?  

The authors also didn't address belowground biodiversity and how that might be coupled.

Still, I think readers in Agronomy will find it useful.  If the authors are willing to expand with more concrete examples and a consideration of belowground diversity, their audience could grow further.

Author Response

Reviewer #1

As a review, this is very nice work.  It captures much of the relevant literature, lays out where we are and provides some thoughts about moving to a more comprehensive understanding of grasslands through coupled biodiversity and biogeochemical modeling.

We thank the reviewer for these comments.

While I think it is publishable as it is, it does fall short in that it provides very little detail on exactly what biogeochemical processes can be coupled to biodiversity or how that can be done with specific examples.  The authors lists some processes and services, but they didn't create any concrete examples.  For instance, what might a biogeochemical model of belowground N transformations look like when coupled to plant diversity?  

The authors also didn't address belowground biodiversity and how that might be coupled.

Still, I think readers in Agronomy will find it useful.  If the authors are willing to expand with more concrete examples and a consideration of belowground diversity, their audience could grow further.

The reviewer suggests that we add detail about model implementation and more consideration of belowground biodiversity. However, our review shows that there is no unique way of implementing the interaction between biodiversity and biogeochemistry, primarily because the mechanisms have not been elucidated sufficiently. We thus decided to recommend parallel development of different modelling approaches in our roadmap (5.3). In the first two of these approaches (presented in 4.1 and 4.2), competition is not explicitly being simulated and the representation of biogeochemistry is not changed from the original model – so implementation is as straightforward as the scheme presented in Fig. 6. We added the following sentences to 4.2 to emphasize the conservation of model structure in this case: “In fact, in this approach model structure regarding biogeochemistry is not changed at all: the same processes are simulated as before. However, the calculation of process rates will now depend on the relative abundances of the different plant species or PFTs where their parameter values differ. In short, the impact of biodiversity becomes an emergent property of community dynamics.” The situation is different for the third approach (4.3) which does involve explicit simulation of competition for finite resources, but for that approach we do explain that different choices are possible depending on which resources are considered in the model.

            We agree with the reviewer as to the importance of belowground biodiversity, and mention this briefly in sections 1.1, 2.1 and 2.3. In the latter section, we indicate that changes in plant diversity may operate through co-evolving soil microbial composition. However, as explained in sections 1.2 and 1.3, our primary focus in this paper is on higher plant diversity, which is under more direct control of management. We therefore prefer to keep discussion on soil biodiversity brief, but we do now mention soil microbial biomass as a possible component of BGMs in 3.1, and we added the following short text to 4.3: “However, BGMs exist that simulate the impact of litter substrate quality on soil microbial biomass and the impact of the microbial biomass on organic matter decomposition and nutrient availability [79]. Such models could be expanded with the capability to not only simulate microbial biomass but also – in an aggregated way - microbial biodiversity which is known to vary with plant species diversity through differences in litter quality [49] and which in turn affects nutrient availability to the plants [73].”

Reviewer 2 Report

The manuscript by van Hoijen et al. reviews the role of biodiversity in ecosystem service provision in grasslands, and modelling approaches to the describe such a role. This topic is timely and suitable for the journal Agronomy. Recent articles cited in this review cover similar topics (e.g., Van Hoijen et al., 2018, Agronomy), but with a different angle, so this contribution is sufficiently novel – however, I have several suggestions to increase novelty and better fill the knowledge gap this work identifies. Moreover, I am under the impression that this work could benefit from a more specific link between processes and models, as described in my main comment below. As a result, this is a well-written, but somewhat qualitative review that misses the opportunity to make specific statements on which processes current models lack to be able to describe biodiversity effects on grassland ES.

Main comment

This work lacks a clear connection between the initial sections on the role of grassland biodiversity and the later sections on modelling biodiversity, which should be the main focus of this review according to the title and abstract. Sections 2.1 and 2.2 are informative, but how is this information connected to models? If these sections are meant to provide motivation for modelling biodiversity, I would move material from here to the Introduction. I would rather expect at this point in the manuscript a summary of the mechanisms that drive biodiversity effects on ES. Models (or at least biogeochemical and ecological models mentioned here) are based on our mechanistic understanding of plant and ecosystem functioning, so if the focus is on modelling biodiversity for ES prediction, it would be good to point out through which processes biodiversity influences ES. I realize many of these processes are unknown, but not all. Then one can argue whether models do include the necessary processes (i.e., are structurally sound) for this aim, or whether they need some ‘improvement’ in the most general sense (i.e., which processes are missing). Some mechanisms are only very briefly presented on P7, but more are at play that are not mentioned or discussed in detail; e.g., exploitation of different niches in space and time allowing better resource use, negative plant-plant interactions via shading or competition for soil nutrients, mutualistic interactions (?). Which of these mechanisms/processes/ecological hypotheses are included in biogeochemical or ecological models? A table summarizing which among the cited models are structurally suitable for describing biodiversity effects on ES would be useful to make the point that BGC models need to be informed by ecological models to advance this field.

Other comments (not easy to point to a specific position in the text without line numbers…)

Abstract: “uncertainty about the quality of predictions” is mentioned, but in my view the issue is more lack of resolution – many biogeochemical models do not have equations describing a diverse plant (or microbial) community. Figure 1: this figure is not very informative and I would suggest removing it. Section 1.1: pollinators are mentioned, but then why not mentioning the role of plant diversity to provide habitat for other faunal groups? P4, top paragraph: which spatial patterns of grassland ES is this paragraph referring to? At which scale – field, landscape, regional? The statement that most BGMs neglect plant diversity is a bit too strong, as many do include at least descriptions of plant functional types. “Higher plant species” needs a clarification or rephrasing – I suppose the authors are narrowing down the scope by defining diversity as number of plant species, genotypes, or phenotypes. P4, second paragraph, last line: “model improvements” is a bit vague – perhaps better to write about integration of modeling approaches, and explain what type of improvements are needed (more/different state variables, new concepts/theories)? P4, third paragraph: not clear what “positioning paper” means. P4, last paragraph: the text “BGMs are generally… Ma et al. [37]).” seems to belong to section 1.2. P5, third bullet point on top: at this point, it is not clear why couldn’t ecological models be modified instead (it will become clearer later). Figure 3: this figure could benefit from an extra panel with similar structure to describe drought responses of plant communities and ANPP. There should be enough information in the literature to draft such a schematic, which would support the numerous statements about drought effects being mitigated by plant diversity in grasslands. P9, end of section 3.1: earlier works by Thornley et al. (e.g., 1995, Annals of Botany 75) already included different plant functional types and their interactions in the context of a biogeochemical model. P9, section 3.2, second line: rather than “small models” I would suggest “minimal models” or “minimalist models”. Section 3.3, title: I might be wrong, but my impression is that models described here are not “for biodiversity”, but rather are meant to represent to some degree plant community dynamics with an effect on ES. Figure 5: what do “(here)” and “(elsewhere)” mean? Pollinators are not really considered in this work… P11, end of first bullet point: in many if not most models, acclimation and trait changes due to evolutionary dynamics are not described, so I would be hesitant to state that traits are dynamic in most BGMs. P11, second bullet point: there are several analytical BGMs, for example those developed by Cherif and Loreau (e.g., 2013, Proc. R. Soc. B 280, and references therein) – perhaps they do not describe biodiversity, but they are analytical BGMs, so some clarifications and distinctions are needed. Section 4.1: the title points to a dynamic metric for biodiversity, but this section is more about a constant metric, so I would suggest adjusting the title or expanding on the idea of a dynamic metric. This addition would also clarify what is suggested in the sentence “The biodiversity-parameter(s) could then…”, which is now a bit unclear/speculative. P12, section 4.2: is the term “states” referring to “state variables” in biogeochemical models? P13, second paragraph, third line: suggested re-phrase “number of state variables and rates”. Section 5.2: the ideas explained at the end of P15 are interesting, but I wonder if a cautionary note could be added – how traits are optimized might depend on the presence of competitors that push optimal behaviors in different directions compared to what would happen in a monoculture. In other words, the presence of other species changes the constraint landscape in which optimal behavior is calculated. P17, first bullet point under “Hybrid mechanistic-probabilistic modelling”: I do not see anything related to probabilistic modelling in this bullet point.

Author Response

Reviewer #2

The manuscript by van Hoijen et al. reviews the role of biodiversity in ecosystem service provision in grasslands, and modelling approaches to the describe such a role. This topic is timely and suitable for the journal Agronomy. Recent articles cited in this review cover similar topics (e.g., Van Hoijen et al., 2018, Agronomy), but with a different angle, so this contribution is sufficiently novel

We thank the reviewer for the positive and constructive comments.

– however, I have several suggestions to increase novelty and better fill the knowledge gap this work identifies. Moreover, I am under the impression that this work could benefit from a more specific link between processes and models, as described in my main comment below. As a result, this is a well-written, but somewhat qualitative review that misses the opportunity to make specific statements on which processes current models lack to be able to describe biodiversity effects on grassland ES.

Main comment

This work lacks a clear connection between the initial sections on the role of grassland biodiversity and the later sections on modelling biodiversity, which should be the main focus of this review according to the title and abstract. Sections 2.1 and 2.2 are informative, but how is this information connected to models? If these sections are meant to provide motivation for modelling biodiversity, I would move material from here to the Introduction. I would rather expect at this point in the manuscript a summary of the mechanisms that drive biodiversity effects on ES. Models (or at least biogeochemical and ecological models mentioned here) are based on our mechanistic understanding of plant and ecosystem functioning, so if the focus is on modelling biodiversity for ES prediction, it would be good to point out through which processes biodiversity influences ES. I realize many of these processes are unknown, but not all. Then one can argue whether models do include the necessary processes (i.e., are structurally sound) for this aim, or whether they need some ‘improvement’ in the most general sense (i.e., which processes are missing). Some mechanisms are only very briefly presented on P7, but more are at play that are not mentioned or discussed in detail; e.g., exploitation of different niches in space and time allowing better resource use, negative plant-plant interactions via shading or competition for soil nutrients, mutualistic interactions (?). Which of these mechanisms/processes/ecological hypotheses are included in biogeochemical or ecological models? A table summarizing which among the cited models are structurally suitable for describing biodiversity effects on ES would be useful to make the point that BGC models need to be informed by ecological models to advance this field.

This comment asks for a closer connection between Section 2 and the later sections of the paper where we propose approaches to model improvement. The reviewer focuses on process representation in BGMs. We partly answered this question in our response to Reviewer 1, where we explained that our review shows that model improvement is not a simple matter of process identification and representation. First of all, knowing about the existence of a process is not enough, there must also be robust understanding of how the processes are controlled to allow quantitative modelling. But such mechanistic understanding of the impacts of biodiversity is missing, as we state in 1.1 (“While there is mounting evidence that biodiversity increases the stability of ecosystem processes in changing environments, the mechanisms that underlie these effects in grasslands are still poorly understood [27]”), and reiterate in 2.3 (“The impacts of biodiversity must be realized through the ecophysiological processes in the grassland, but the key mechanisms have generally not been identified in great detail [50]”). We therefore conclude in 5.1 that “Despite the availability of manipulation studies, our understanding of the biotic and abiotic controls on grassland processes remains incomplete. When reviewing the results of the 15-year Jena grassland experiment, Weisser and colleagues [50] concluded that the mechanisms underlying observed positive effects of biodiversity on grassland ES are not well known, and that likely several mechanisms are acting simultaneously. This lack of understanding hampers the identification of optimal model structure”.

            The reviewer is correct in stating that a primary purpose of Section 2 was to motivate the need for modelling biodiversity in BGMs. And our review in Section 2 indeed revealed that biodiversity plays an essential role in modulating the impact of environmental change on ES, so BGMs that ignore biodiversity need to be improved. Section 2 thus establishes the ‘why’ and the following sections discuss the ‘how’. To emphasize the motivational role of Section 2 we changed its title to “Why consider biodiversity in grassland models?” And we begin the study of how models can be improved in Section 3 by explaining the idiosyncrasies of BGMs and ecological biodiversity modelling. That culminates in 3.3, which shows the incompatibilities between modelling paradigms (of which the choice of variables – including process representation - is only one), and we suggest ways forward in Sections 4 and 5.

            There is an additional important link between Section 2 and the remainder of the paper, which refers to Section 2.3. We show in that section that not only mechanistic understanding is limited, but also that empirical response functions are often inconsistent among studies, in particular when comparing biodiversity manipulation studies and non-intrusive observational studies. The need to explain that apparent mismatch provides another strong motivation for developing BGMs that represent biodiversity, and the potentially explanatory role of modelling is indicated in Fig. 4. That figure is developed in Section 2 and is referred to prominently in Sections 5.1 and 5.3 (the roadmap). Resolving the mismatch will require research where different modelling approaches that combine biogeochemistry and biodiversity are compared to benchmark data, as we propose in our roadmap.

Abstract: “uncertainty about the quality of predictions” is mentioned, but in my view the issue is more lack of resolution – many biogeochemical models do not have equations describing a diverse plant (or microbial) community.

We agree that it is the absence of equations for biodiversity dynamics that affects the reliability of the BGMs, but that is what we had wanted to convey in the first place. We replaced the phrase “mostly remains disjoint from” with the perhaps clearer “is generally not coupled to”.

Figure 1: this figure is not very informative and I would suggest removing it.

We prefer to keep this figure in to focus the mind of the reader from the beginning on the role of biodiversity as a possible modulator of environmental impacts on ES. Especially in the community of biogeochemistry modellers, this is not the common perspective. The presence of Fig. 1 allows us to write in Section 2.3 that the analysis by Liu et al. shown in Fig. 3 is “rare example where the general scheme of Fig. 1 was clarified in detail”.

Section 1.1: pollinators are mentioned, but then why not mentioning the role of plant diversity to provide habitat for other faunal groups?

We mentioned pollinators as an example of a faunal group that provides a specific ES. However, we agree with the reviewer that other faunal groups can provide ES as well (e.g. earthworms and natural enemies of pests and diseases), so we changed the text and modified Fig. 5 accordingly. This also improves the consistency with our text further on in the Introduction “The importance of plant biodiversity is emphasized because it is closely linked to soil health and functioning, due to its benefits for soil microorganisms and other fauna [15]”.

P4, top paragraph: which spatial patterns of grassland ES is this paragraph referring to? At which scale – field, landscape, regional?

We replaced the strong term “spatial pattern” with the less suggestive “spatial variation”. In this early part of the paper we have not yet described BGMs in detail (that is done in 3.1) so we leave the explanation that BGMs are point-support models that can be applied to various scales depending on the spatial resolution and extent of the available information about driver variables to 3.1.

The statement that most BGMs neglect plant diversity is a bit too strong, as many do include at least descriptions of plant functional types. “Higher plant species” needs a clarification or rephrasing – I suppose the authors are narrowing down the scope by defining diversity as number of plant species, genotypes, or phenotypes.

We agree with this comment and have softened the language. And we indeed narrowed the scope of the paper by focusing on higher plant diversity with only minor attention to microbial diversity (see our reply to Reviewer 1 for more details).

P4, second paragraph, last line: “model improvements” is a bit vague – perhaps better to write about integration of modeling approaches, and explain what type of improvements are needed (more/different state variables, new concepts/theories)?

The statement about model improvements being needed is indeed vague, but that was deliberate, as it sets the scene for what we aim to provide in this paper. To clarify this goal of the paper, we rephrased the sentence to: “model improvements are needed, which we aim to elucidate in this paper”.

P4, third paragraph: not clear what “positioning paper” means.

We changed the incorrect term to “position paper”.

P4, last paragraph: the text “BGMs are generally… Ma et al. [37]).” seems to belong to section 1.2.

This text about BGMs not including biodiversity in any detail indeed belongs to the topic of 1.2, but similar statements were made there already. To avoid redundancy, we removed most of the text.

P5, third bullet point on top: at this point, it is not clear why couldn’t ecological models be modified instead (it will become clearer later).

Yes, the reasons for extending BGMs with biodiversity rather than the other way around are indeed clarified in later sections (including 5.2). We chose to keep the bullet points at the end of the Introduction brief and matter-of-fact to summarise what the reader can expect from the paper. Further on in the text, after the model types are presented and compared, the path forward is presented in more detail.

Figure 3: this figure could benefit from an extra panel with similar structure to describe drought responses of plant communities and ANPP. There should be enough information in the literature to draft such a schematic, which would support the numerous statements about drought effects being mitigated by plant diversity in grasslands.

We did not opt to do this, because no rigorous causal analysis as performed by Liu et al. (2019) has been carried out for drought impacts. The causal analysis by Liu et al. that is depicted in Fig. 3 is in fact a rare example where mechanistic knowledge on causal pathways from environment through biodiversity to ES has been quantified (although their analysis still does not specify process detail), whose uniqueness and importance we wanted to emphasize. We now explain in the legend of Fig. 3 that it represents an “example of causal relationships” to avoid the impression that it captures the system as a whole.

P9, end of section 3.1: earlier works by Thornley et al. (e.g., 1995, Annals of Botany 75) already included different plant functional types and their interactions in the context of a biogeochemical model.

We refer to Thornley’s work on grass-legume modelling elsewhere in the paper (4.2).

P9, section 3.2, second line: rather than “small models” I would suggest “minimal models” or “minimalist models”.

We followed the first suggestion of the reviewer.

Section 3.3, title: I might be wrong, but my impression is that models described here are not “for biodiversity”, but rather are meant to represent to some degree plant community dynamics with an effect on ES.

We modified the title to “… models for biodiversity-ES relations”.

Figure 5: what do “(here)” and “(elsewhere)” mean? Pollinators are not really considered in this work…

These terms were meant to indicate that the presence of pollinators at the field of interest (“here”) should be distinguished from how the biogeochemistry-biodiversity interaction at that field affects the presence of pollinators “elsewhere”. However, we agree that none of this was very clear, and we have modified the figure (see also our response to the earlier comment by the reviewer regarding pollinators vs. other faunal groups).

P11, end of first bullet point: in many if not most models, acclimation and trait changes due to evolutionary dynamics are not described, so I would be hesitant to state that traits are dynamic in most BGMs.

We agree with this statement, but we do not write that traits are dynamic in most BGMs. We refer to them as generally being treated as “static inputs (i.e. model parameters)”.

 “P11, second bullet point: there are several analytical BGMs, for example those developed by Cherif and Loreau (e.g., 2013, Proc. R. Soc. B 280, and references therein) – perhaps they do not describe biodiversity, but they are analytical BGMs, so some clarifications and distinctions are needed.

The reviewer is correct in that analytical equilibrium analysis of the Cherif & Loreau class of models is possible if boundary conditions are fixed. But the dynamics – as opposed to the equilibria - of these models can be investigated only locally, by analysis of the derivatives in the Jacobian matrix, so the state variables cannot be expressed as analytical functions of time. These models thus require numerical solution by computer to analyse dynamics. In our paper we define BGMs as dynamic process-based models with driving variables such as weather and N-deposition (which are boundary conditions) that vary over time, so that would exclude the biogeochemistry modelling for equilibrium analysis. However, we did change our text to allow for the less general approaches.

Section 4.1: the title points to a dynamic metric for biodiversity, but this section is more about a constant metric, so I would suggest adjusting the title or expanding on the idea of a dynamic metric. This addition would also clarify what is suggested in the sentence “The biodiversity-parameter(s) could then…”, which is now a bit unclear/speculative.

We agree with this comment – our text is mainly about a constant metric - and we have amended the title.

P12, section 4.2: is the term “states” referring to “state variables” in biogeochemical models?

Yes it is, and we clarified this by writing “states of the BGM”.

P13, second paragraph, third line: suggested re-phrase “number of state variables and rates”.

We made this change.

Section 5.2: the ideas explained at the end of P15 are interesting, but I wonder if a cautionary note could be added – how traits are optimized might depend on the presence of competitors that push optimal behaviors in different directions compared to what would happen in a monoculture. In other words, the presence of other species changes the constraint landscape in which optimal behavior is calculated.

We thank the reviewer for the positive words and we included the suggested cautionary note in the following text: “Combining coordination of photosynthesis and the leaf economics spectrum should thus allow three major plant traits to vary in the model directly with environmental conditions, although the extent to which the presence of competitors may change the location of the optimum for each species needs to be ascertained”.

17, first bullet point under “Hybrid mechanistic-probabilistic modelling”: I do not see anything related to probabilistic modelling in this bullet point.

We clarified the probabilistic nature of adding empirical relationships to the outputs from the BGM as follows: “These added relationships are not mechanistic and may not be robust, but they can be implemented as conditional probability distributions to account for uncertainty.” They are thus similar to ‘observational operators’ in weather and climate forecasting which allow predictions of state variables to be compared to observations.

Reviewer 3 Report

1) I would prefer the abbreviations in the beginning. As it is now it's difficult to follow the text. Also have an explanation of the abbreviation in the text. It's in some cases but not all.

2) In abbreviations, you have SLA. What is that? It just stand alone.

3) Maybe is it a rule of the magazine, but I would have prefered to read about the figure before I see it, now is it the opposite way.

Author Response

Reviewer #3

1) I would prefer the abbreviations in the beginning. As it is now it's difficult to follow the text. Also have an explanation of the abbreviation in the text. It's in some cases but not all.

2) In abbreviations, you have SLA. What is that? It just stand alone.

3) Maybe is it a rule of the magazine, but I would have prefered to read about the figure before I see it, now is it the opposite way.

We agree with comments 1 and 2, and have corrected the list of abbreviations and moved it to the beginning of the manuscript. We allocated the figures to locations where they do not clutter each other, and believe that they are distributed in a visually appealing way over the manuscript.